# Numerical Analysis of the Activated Combustion High-Velocity Air-Fuel Spraying Process: A Three-Dimensional Simulation with Improved Gas Mixing and Combustion Mode

**DOI:** 10.3390/ma14030657

**Published:** 2021-01-31

**Authors:** Fuqiang Liu, Zhiyong Li, Min Fang, Hua Hou

**Affiliations:** School of Materials Science and Engineering, North University of China, Taiyuan 030051, China; lfq@nuc.edu.cn (F.L.); fangmin@nuc.edu.cn (M.F.); houhua@nuc.edu.cn (H.H.)

**Keywords:** AC-HVAF spraying, porous ceramic sheet, radial inlet, combustion reaction, air-fuel ratio one-step method, two-step method

## Abstract

Owing to its low flame temperature and high airflow velocity, the activated combustion high-velocity air-fuel (AC-HVAF) spraying process has garnered considerable attention in recent years. Analyzing the velocity field, temperature field, and composition of AC-HVAF spray coatings plays a vital role in improving the quality of coatings. In this study, an actual spray gun is adopted as a prototype, and the radial air inlets are introduced to improve the reaction efficiency so that the chemical reaction can be completed in the combustion chamber. Furthermore, a complete three-dimensional (3D) model is established to examine the effects of radial inlets and porous ceramic sheet on the combustion and flow fields. The hexahedral cells are used to discretize the entire model for reducing the influence of false-diffusion on the calculation results. The gas flow field is simulated by the commercial Fluent software, and the results indicate that the porous ceramic sheet effectively reduces the turbulent dissipation of the airflow with a good rectification effect (the ceramic sheet ensures a consistent airflow direction). The radial inlets and the porous ceramic sheet promote the formation of vortex in the combustion chamber, increase the residence time and stroke of the gas in the combustion chamber, and improve the probability of chemical reactions. In addition, it is observed that the stability of velocity for the airflow is strongly related to the airflow density.

## 1. Introduction

The activated combustion high-velocity air-fuel (AC-HVAF) spraying is a novel surface spraying process developed on the basis of high velocity oxygen-fuel (HVOF) spray coating process [1]. Compared with HVOF spraying, AC-HVAF spraying has the advantages of low flame temperature, low cost, and efficient processing [2,3,4,5]. Besides that, AC-HVAF coatings have a lower oxygen content, lower porosity, denser layer, and beneficial compressive stress on the surface as compared with HVOF coatings [6,7,8]. Therefore, AC-HVAF spraying has received considerable attention and can be used to prepare a wide range of coatings, including pure metals, alloys, intermetallic compounds, nano-coatings, and amorphous coatings [2,9,10,11,12,13].

Although several studies have focused on understanding the formation mechanism of AC-HVAF coatings, the high airflow velocity and the difficulty in measuring physical quantities pose a significant challenge for further research. Computer simulation technology has become an effective means to solve this problem. Over the past decades, there have been relatively few simulation studies on HVAF spraying. Gorlach simulated the thermal stress and temperature changes of different materials of sandblasting nozzles on spray gun over time using a two-dimensional (2D) swirl axisymmetric model and provided useful suggestions for the selection of HVAF nozzle materials [14]. To realize cold spraying through HVAF coating, Yuan et al. injected liquid water into the high-velocity airflow through a small radial hole downstream of the throat of the spray gun, which reduced the temperature of the airflow. Further, they established a three-dimensional (3D) model to obtain the optimum mass flow rate of liquid water and the flame characteristics at such a flow rate [15]. Jiang et al. assumed that fuel and air were fully mixed and regarded the small holes on the ceramic sheet in the spray gun as the gas inlets to establish a 3D model. Based on this model, the airflow velocity, temperature, pressure, and other parameters under a certain mass flow rate were obtained, which were used to simulate the flight process of particles [16]. Based on a 2D swirl axisymmetric model, Gao et al. examined the influence of different mass flow rates and nozzle geometries on the flow field of HVAF spraying [17]. In addition, although the structure of spray gun and the spraying principle of other spraying such as HVOF spraying [18,19,20,21,22,23,24,25,26,27,28,29,30,31,32], plasma spraying [33,34,35], and cold spraying [36] are extremely different from HVAF spraying, the related simulation studies also provided a useful reference for this study on AC-HVAF spraying.

The above studies have provided interesting results on AC-HVAF spraying. However, because of their diverse research objectives, different studies used different models, and the actual spray gun was simplified to various degrees, which may lead to errors in the simulated results with respect to the actual situation. In particular, the chemical reaction in the combustion chamber plays a decisive role in the entire flow field. However, the earlier studies ignored the following question: Does the chemical reaction continue within the entire spray gun? Apparently, the degree of chemical reaction determines the heat released in the reaction, which profoundly affects the composition, pressure, temperature, and velocity of the gas. These physical quantities have a significant impact on the spraying process, and they should be different if the end position of chemical reaction is different. Owing to the lack of reliable testing methods and experimental data, it is difficult to determine the position at which the chemical reaction in the actual spray gun is completed. The chemical reaction in the combustion chamber is governed by several factors. Among them, as a porous medium, the ceramic sheet has a considerable effect on the combustion reaction [37,38,39], but its influence mechanism needs to be further examined.

According to the flame characteristics of actual spray guns, the chemical reaction may well be completed inside the spray gun. Therefore, it is assumed in this study that the combustion reaction is completed inside the combustion chamber. Through a large number of simulation experiments, it was found that the combustion reaction may not be completed quickly in an extremely small space in the combustion chamber. This may be attributed to the fact that the existing chemical reaction models are governed by the generation, movement, and dissipation of turbulence [40,41,42]. The actual turbulence might be much more complicated than the simulated turbulence because the smooth wall surface used in the simulation cannot produce an identical vortex as the microscopic uneven surface of the actual spray gun. To resolve the above issues, the porous ceramic sheet is used in the simulation to examine its influence on the combustion, and radial compressed air inlets are added to the wall of the combustion chamber to enhance the vortexes, thereby promoting the chemical reaction. Further, a full 3D model is established to characterize the ceramic sheet and the radial inlets. Four kinds of meshes are used to analyze the influence of the type of mesh cells, ceramic sheet, and radial inlets on the flow field. The commercial software Fluent is used to simulate the combustion and species transportation process, as well as the distribution of physical quantities such as velocity, temperature, and composition. The effect of chemical reactions on the flow field is examined through two chemical reactions (one-step method and two-step method). The present research work tried to provide theoretical reference for understanding the mechanism of the AC-HVAF spraying process by a simulation study.

## 2. Configuration of the Model

### 2.1. Model Structure

The model structure and spraying mechanism are shown in Figure 1. When the spray gun is working, the compressed air and propane are separately injected into the combustion chamber in a discrete form. After ignition of the AC-HVAF gun, the ceramic sheet is preheated up to the auto-ignition temperature of the mixture gas; then, the high temperature ceramic sheet is used as catalytic medium to continuously ignite the mixture, which is called activate combustion. The combustion products enter the rectifying chamber through hundreds of orifices of ceramic sheets and are axially injected with the nitrogen pre-mixed powder particles, which are accelerated in the nozzle to hit the substrate and form a coating.

The model structure mainly includes gas inlets, combustion chamber, rectifying chamber, porous ceramic sheet, and Laval nozzle. The most significant difference between the model and the actual spray gun structure is the additional radial compressed air inlets. All gases are discretely injected into the combustion chamber through 74 small holes (compressed air inlets: 16 in the axial direction, 40 in the radial direction; propane inlets: 18 in the axial direction).

The square holes are evenly distributed on the ceramic sheet, and there is a hollow circular hole with a diameter of 12 mm at the axial center of the combustion chamber and the ceramic sheet, which is the installation position of the powder injector for conveying powder particles and carrier gas (N_2_). The detailed dimensions of the model are shown in Figure 2.

### 2.2. Computational Domain and Boundary Conditions

As shown in Figure 2, the flow field includes internal flow field and external flow field. In the internal flow field, the mass flow inlets are adopted as the gas inlets. Here, the temperatures of the compressed air and propane are assumed to be 320 K and 340 K, respectively (propane is preheated). The wall temperature of the spray gun is assumed to be 600 K due to the cooling effect of the compressed air. As the ceramic sheet has high heat capacity, high wall temperature is employed, which has a tendency of linear increase along the axis from 1500 K to 1800 K.

The external flow field is assumed to be a cylinder with a diameter of 400 mm to ensure the reliability of the simulation results. The pressure outlet is located at the cylindrical surface, and is set to 1 atm. The left boundary is defined as the velocity inlet of the external flow field, and the velocity decreases gradually from the muzzle to the pressure outlet, i.e., the velocity increases with decrease in the distance from the jet center. The maximum and minimum velocities are 8 m/s and 0.1 m/s, respectively. The temperature of the ceramic sheet and the velocity at the velocity inlet of the external flow field change with the position, which is realized by the following custom functions:(1)$$T\left(x\right)=(T_{max}-T_{min})\cdot \frac{x-x_{min}}{x_{max}-x_{min}}+T_{min}$$
(2)$$u\left(r\right)=u_{max}\cdot \left(1-\sqrt{\frac{r-r_{min}}{r_{max}-r_{min}}}\right)$$
$$T_{max}=1800\;K,\;T_{min}=1500\;K,\;x_{max}=24\;\mathrm{mm},\;x_{min}=20\;\mathrm{mm}\;u_{max}=8\;m/s,\;u_{min}=0.1\;m/s,\;r_{max}=200\;\mathrm{mm},\;r_{min}=4\;\mathrm{mm}.$$
where *T* and *x* represent the temperature and the absolute value of the *x*-axis coordinate of each point on the ceramic sheet, respectively; and u and r represent the velocity and the absolute value of the radial coordinate in the left boundary of the outflow field, respectively. The corresponding angular symbols represent the maximum and minimum value, respectively.

### 2.3. Meshing

To examine the effect of the type of mesh cells, ceramic sheet, the size of ceramic holes, and radial inlets on the simulation results, four types of meshes are used, and the detailed parameters are listed in Table 1. Figure 3 shows the layout of the third meshes, which consist of 1,082,096 hexahedral cells and 1,149,140 nodes, and the mesh independence is tested. As mentioned earlier, the small holes in the model such as gas inlets and the ceramic holes are considered to be square holes (different from the round holes in the actual situation) for improving the quality of the meshes and reducing the number of meshes. Although the geometry is not completely consistent with the actual spray gun, it is verified that there is almost no difference with the simulation results based on round holes.

### 2.4. Flow Dynamic Model

The AC-HVAF spraying process is accompanied with complex physical, chemical, metallurgical, and species transport processes, where the gas composition, momentum, and energy transport are the primary difficult points of simulation. From the perspective of fluid mechanics, as a continuous phase, gas follows the conservation of mass, momentum, and energy. Therefore, the governing equations of gas flow include mass conservation equation, momentum conservation equation, energy transport equation, and species transport equation [43,44].

(1)Mass conservation equation:(3)$$\frac{\partial \rho }{\partial t}+\nabla \cdot \left(\rho \vec{v}\right)=0$$(2)Momentum conservation equation:(4)$$\frac{\partial }{\partial t}\left(\rho \vec{v}\right)+\nabla \cdot \left(\rho \vec{v}\vec{v}\right)=-\nabla p+\nabla \cdot \left({\overset{̿}{\tau }}_{eff}\right)+\nabla \cdot \left(-\bar{\rho v^{\prime }v^{\prime }}\right)$$

The stress tensor ${\overset{̿}{\tau }}_{eff}$ is given by
$${\overset{̿}{\tau }}_{eff}=\mu_{eff}\left[\left(\nabla \vec{v}+\nabla {\vec{v}}^{T}\right)-\frac{2}{3}\nabla \vec{v}I\right]$$

(3)Energy transport equation:(5)$$\frac{\partial }{\partial t}\left(\rho E\right)+\nabla \cdot \left[\vec{v}\left(\rho E+p\right)\right]=\nabla \cdot \left[k_{eff}\nabla T-\underset{j}{\sum }h_{i}{\vec{J}}_{i}+\left({\overset{̿}{\tau }}_{eff}\cdot \vec{v}\right)\right]+S_{h}$$(4)Species transport equations:

(6)$$\frac{\partial }{\partial t}\left(\rho Y_{i}\right)+\nabla \cdot \left(\rho \vec{v}Y_{i}\right)=-\nabla \cdot {\vec{J}}_{i}+R_{i}$$
where ρ is the density of airflow, p is the static pressure, $\vec{v}$ is the velocity, $-\bar{\rho v^{\prime }v^{\prime }}$ is the Reynolds stress, $\mu_{eff}$ is the effective viscosity, *I* is the unit tensor, *E* is the enthalpy, $k_{eff}$ is the effective conductivity, *T* is the temperature, ${\vec{J}}_{i}$ is the diffusion flux of species i, $h_{i}$ is the sensible enthalpy of species i, $S_{h}$ is the heat generation rate from chemical reactions, $Y_{i}$ is the mass fraction of species i, and $R_{i}$ is the net rate of production of species i by chemical reaction.

The realizable *k-ε* model based on the standard *k-ε* model is selected to characterize the turbulent motion of airflow, where *k* is the turbulent kinetic energy and ε is the rate of dissipation of turbulent energy. The realizable *k-ε* model modifies the standard *k-ε* model in two important ways: it contains an alternative formulation for the turbulent viscosity; and a modified transport equation for the dissipation rate, ε, has been derived from an exact equation for the transport of the mean-square vorticity fluctuation [45]. Therefore, compared with the standard *k-ε* model, the realizable *k-ε* model has advantages in the following simulation: free flows including jets and mixing layers, channel and boundary layer flows, and separated flows. Especially noteworthy is the fact that the realizable *k-ε* model resolves the round-jet anomaly [46,47,48,49]. The local values of *k* and *ε* are obtained by solving the following equations [45]:(7)$$\frac{\partial }{\partial t}\left(\rho k\right)+\frac{\partial }{\partial X_{j}}\left(\rho ku_{j}\right)=\frac{\partial }{\partial X_{j}}\left[\left(\mu +\frac{\mu_{t}}{\sigma_{k}}\right)\frac{\partial k}{\partial X_{j}}\right]+G_{k}+G_{b}-\rho \varepsilon -Y_{M}+S_{k}\;$$
(8)$$\frac{\partial }{\partial t}\left(\rho \varepsilon \right)+\frac{\partial }{\partial X_{j}}\left(\rho \varepsilon u_{j}\right)=\frac{\partial }{\partial X_{j}}\left[\left(\mu +\frac{\mu_{t}}{\sigma_{\varepsilon }}\right)\frac{\partial k}{\partial X_{j}}\right]+\rho C_{1}S\varepsilon -\rho C_{2}\frac{\varepsilon^{2}}{k+\sqrt{v\varepsilon }}+C_{1\varepsilon }\frac{\varepsilon }{k}C_{3\varepsilon }G_{b}+S_{\varepsilon }$$
(9)$$G_{b}=\beta g_{i}\frac{\mu_{t}}{Pr_{t}}\frac{\partial T}{\partial x_{i}}$$
$$\beta =-\frac{1}{\rho }{\left(\frac{\partial \rho }{\partial T}\right)}_{p}$$

For ideal gases, Equation (9) reduces to
(10)$$G_{b}=-g_{i}\frac{\mu_{t}}{\rho Pr_{t}}\frac{\partial T}{\partial x_{i}}$$
$$C_{1}=\max\left[0.43,\frac{\eta }{\eta +5}\right],\;\eta =S\frac{k}{\varepsilon },\;S=\sqrt{2S_{ij}S_{ij}},\;Pr_{t}=0.85$$

Here, $G_{k}$ represents the generation of turbulence kinetic energy due to the mean velocity gradients. $G_{b}$ is the generation of turbulence kinetic energy due to buoyancy. $Y_{M}$ represents the contribution of the fluctuating dilatation in compressible turbulence to the overall dissipation rate. $Pr_{t}$ is the turbulent Prandtl number for energy and $g_{i}$ is the component of the gravitational vector in the *i*th direction. $C_{2}$ and $C_{1\varepsilon }$ are constants. $\sigma_{k}$ and $\sigma_{\varepsilon }$ are the turbulent Prandtl numbers for k and ε, respectively. $S_{k}$ and $S_{\varepsilon }$ are user-defined source terms.

### 2.5. Combustion Model

The Eddy dissipation model (EDM) is used for the combustion reaction between air and propane. In this model, it is assumed that the rate of chemical reaction is higher than that of the turbulent mixing, which implies that the reaction progress is determined by the mixing rate between fuel and oxidant. In fact, the reaction process of propane and oxygen is rather complicated, which involves much intermediate products such as CO and NO_x_. However, according to the earlier reported simulation results [16], the concentration of other intermediate products in the full flow field is extremely low, except for CO. To simplify the calculation, the reaction between propane and oxygen is simulated by the one-step method and two-step method [50].

One-step method:(11)$${\;C}_{3}H_{8}+5O_{2}\to 3{\mathrm{CO}}_{2}+4H_{2}O$$

Two-step method:(12)$$2C_{3}H_{8}+7O_{2}\to 6\mathrm{CO}+8H_{2}O$$
(13)$$2\mathrm{CO}+O_{2}\to 2{\mathrm{CO}}_{2}$$

Theoretically, three situations may occur in the chemical reaction: excess air, excess fuel, and stoichiometric numbers that meet the reaction formula. Excess air can cause oxidation of the metal parts of the spray gun and particles; hence, this situation need not be discussed as it has little practical significance. The slight excess fuel has certain advantages because the intermediate product CO can prevent the oxidation of particles.

To analyze the influences of two chemical reactions on the flow field, the parameters listed in Table 2 are examined, and the other parameters remain constant. In the experiment name, the Roman numerals (e.g., I in I-9.5) denote the chemical reaction mode, and the following numbers (e.g., 9.5 in I-9.5) denote the mass flow ratio of air to fuel.

## 3. Results and Discussion

### 3.1. Influence of Mesh Cells Pattern on Simulations

Figure 4 shows the temperature fields of the combustion chamber and the substrate formed using different mesh cells. Comparing Figure 4a,c it can be seen that the temperature field obtained using the fourth meshes has better symmetry than that using the first meshes. After comprehensive comparison of Figure 4a-d, it becomes clear that the asymmetry of temperature field in the combustion chamber affects the subsequent flow field when the first meshes are adopted, and this effect decreases with the large-gradient change in the subsequent flow field. In fact, the other physical quantities such as velocity and composition in the combustion chamber are also distributed asymmetrically when the first meshes are adopted.

Obviously, the model is structurally symmetrical, and the flow field should also be at least somewhat symmetrical. Hence, compared with the hexahedral cells, the simulation results exhibit more errors when tetrahedral cells are adopted. The fourth meshes are structured meshes, whose direction is highly consistent with the fluid flow direction, while tetrahedral cells of the first meshes cannot be aligned with the flow direction. The inconsistency between the meshes’ direction and the flow direction leads to the false-diffusion, which affects the accuracy of the simulation results. The false-diffusion is small when low-velocity flow is simulated, which has a minor impact on the simulation results, but its effect cannot be ignored if the flow rate increases [51]. The airflow velocity of AC-HVAF spraying is rather high, especially the supersonic airflow behind the nozzle throat. Therefore, the false-diffusion caused by hexahedral mesh cells is smaller than that caused by tetrahedral mesh cells in the supersonic flow field, and the former provides higher simulation accuracy.

### 3.2. Effect of Ceramic Sheet on Gas Flow Field

Figure 5 and Figure 6 show the comparisons of velocity and temperature with and without a ceramic sheet in the case wherein the other parameters are the same. It is clear that the ceramic sheet has a rectification effect on the airflow. The velocity and temperature of the airflow through the ceramic sheet are more uniform. Further, it can be seen in Figure 5 that the ceramic sheet increases the vortexes in the combustion chamber and prevents the fluid element with high radial velocity from passing, which increases the residence time and stroke of these fluid elements in the combustion chamber as well as the reaction probability between the fuel and the oxidant. This reduces the production of harmful gases such as CO and NO_x_ that are discharged from the gun, which leads to efficient combustion and reduces the energy consumption.

### 3.3. Effect of Hole Sizes of Ceramic Sheet on Gas Flow Field

The turbulent kinetic energy ***k*** is given by the following [52]:(14)$$k=\frac{1}{2}\left(\vec{{u^{\prime }}^{2}}+\vec{{v^{\prime }}^{2}}+\vec{{w^{\prime }}^{2}}\right)$$

In Equation (14), the right-hand side of the equation represents one half of the sum of velocity variances. The greater the turbulent kinetic energy, the greater the velocity inhomogeneity, resulting in more energy consumption between the airflows.

Figure 7 shows the difference of the turbulent kinetic energy inside the spray gun of three different configurations. As shown in this figure, the ceramic sheet effectively reduces the turbulent kinetic energy inside the spray gun, and the smaller the holes in the ceramic sheet, the smaller the turbulent kinetic energy after the region of the ceramic sheet. The lower turbulent kinetic energy of the flow is conducive to reducing the internal kinetic energy consumption of the airflow, so the higher airflow velocity appears at the muzzle, which is demonstrated by the curves in Figure 8b.

### 3.4. Effect of the Radial Air Inlets on Gas Flow Field

The flow field of the model is completely different if there are no radial inlets. The experimental results in Figure 8 are obtained according to I-9.5. In Figure 8a, the radial inlets are closed and their corresponding mass flow is added to the axial air inlets. As shown in the figure, the combustion reaction occurs over larger area when there are no radial inlets, which results in a more uneven temperature field inside the gun. Figure 8b shows that the airflow velocity is related to both the ceramic sheet and the radial inlets. When there are radial inlets, the smaller the hole in the ceramic sheet, the higher the jet velocity of the gun; however, when there are no radial inlets, the jet velocity of the gun is lower even if there is a ceramic sheet.

It is noteworthy that the radial air inlets are close to the ceramic sheet in the all models; the low-temperature compressed air that enters the combustion chamber is heated by the high-temperature ceramic sheet, and its temperature rises rapidly, which is extremely beneficial for promoting chemical reactions. Combining Figure 5, Figure 6, Figure 7 and Figure 8, it is clear that the gas from radial air inlets increases the radial velocity of the fluid element in front of the ceramic sheet, so they have a low probability of passing through the ceramic sheet. In other words, the radial inlets not only promote combustion, but also enhance the rectification effect of the ceramic sheet. As mentioned above, this can promote the consistency between airflow direction and meshes’ direction, reduce the false-diffusion, and improve the accuracy of the simulation results.

### 3.5. Influence of Different Chemical Reactions on the Simulation Results

#### 3.5.1. Influence of the Different Chemical Reactions on Velocity Field

The velocity of the airflow is the most important indicator of AC-HVAF spraying, and it determines the velocity and heated history of the particles. For the convenience of discussion, the flow field is divided into three zones along the axial direction: zone I indicates the internal flow field, zone II represents the shock zone, and zone III is the region from the end position of shock wave to the substrate. Figure 9 shows an example of these three zones, and the partial contours of the airflow velocity under eight experimental schemes. Table 3 lists the maximum velocity for these experimental schemes. Figure 10 clearly shows the variation in velocity along the axis. Here, it is clear that the velocity change process of each experimental scheme is basically the same.

In zone I, the airflow in the nozzle continues to accelerate and reaches the maximum velocity until the outlet. As shown in Figure 10 and Table 2, the velocities of different experimental schemes have a negligible difference in zone I, which indicates that the different chemical reactions have a minor effect on the velocity in the internal flow field.

In zone II, the airflow is in an over-expanded state, and some diamond shock waves are formed because the pressure at the outlet is lower than the ambient pressure. Because of the impact of the shock waves, the velocity increases and decreases periodically. The momentum exchange between the jet and the surrounding air leads to a gradual decrease in the average velocity and the amplitude of the airflow. The difference is observed at the end positions of the shock zone only between I-9.5 and II-9.5. The end position of the shock zone for I-9.5 is later than that of II-9.5. This indicates that the airflow of I-9.5 has more stability of the velocity, and its high velocity can be maintained for a long distance.

In zone III, the velocity of each experiment drops rapidly, and the difference between them lies in decline rate of the velocity. However, there is no obvious difference between the two chemical reactions.

#### 3.5.2. Influence of the Different Chemical Reactions on Temperature Field

Temperature field is another important factor affecting the coating quality. The temperature distribution near the axis of the spray gun governs the heated history of the particles. However, the temperature field near the substrate influences the final quality of the coating deeply.

Figure 11 shows the change of airflow temperature along the axis. The negligible difference in temperature between different chemical reactions can be seen in all zones except for I-9.5 and II-9.5. However, the differences between them are limited to zone III. As shown in Figure 11, the temperature fields near substrates in zone III show an obvious difference in different experiments, which is mainly attributed to two factors: different degrees of compression effect and released heat by the combustion between the remaining fuel and the oxygen from the external. Figure 12 provides a good proof of this phenomenon.

#### 3.5.3. Influence of the Different Chemical Reactions on Gas Species Transport

Figure 12a,b show that the degree of fuel excess of each experiment, and Figure 12c implies that the chemical reaction basically ends in the combustion chamber, which satisfies the previous assumption. Obviously, the reaction products of different chemical reactions are different because of the excessive fuel. If the two-step method is used, the excess C_3_H_8_ generates CO, so the average density of the gas is lower than the one-step method. The higher the airflow density, the higher the momentum at the same velocity, and the state of the airflow motion is less susceptible to interference by the air from the external flow field. Therefore, the difference in the density of the airflow causes a significant velocity variation after the shock wave, which is the reason that the airflow of I-9.5 has the more stability of the velocity than the one of II-9.5.

Figure 12c shows that the oxygen from the external field rapidly enters the center of the airflow through diffusion and results in a rapid increase of the jet oxygen content. This may lead to the oxidation of the metal powder and increase of the oxygen content in the coating, and decrease the quality of the coating. However, this can only occur when the gas inlets flow rate is insufficient, such as I-15.3-S and II-15.3-S. It should be noted that the combustion reaction of excess fuel is very violent near the substrate because the oxygen from the external field is carried by the jet and concentrates in a small area. Therefore, the temperature of the gas can rise rapidly near the substrate, which will cause the substrate to form a very large temperature gradient (e.g., up to 200 K/mm for I-11.2), as shown in Figure 11.

According to the stoichiometry in the chemical reaction equation, in order to fulfill a complete combustion process, the mass flow ratio of air to fuel should be about 15.6. The simulation results of the two chemical reactions are relatively close when the degree of fuel excess is less than 24% (corresponding to I-11.2 and II-11.2). However, the simulation results were significantly different when the degree of fuel excess was 68% (corresponding to I-9.8 and II-9.8).

In a word, on the premise that the combustion can end in the combustion chamber, the influence of chemical reaction method on the simulation results is significant only when the fuel excess degree is large.

## 4. Conclusions

In this study, a complete 3D model was established using an actual spray gun as a prototype. It is assumed that the combustion reaction is completed inside the combustion chamber. The influence of the mesh cells’ type on the simulation results was analyzed, and the effects of porous ceramic sheet, radial inlets, and chemical reaction process were thoroughly investigated. The main results of the study are summarized as follows:The false-diffusion caused by hexahedral mesh cells is smaller than that caused by tetrahedral mesh cells in the supersonic flow field, and the former provides higher simulation accuracy.The porous ceramic sheet has a rectification effect. The radial inlet and porous ceramic sheets increase the residence time and stroke of the gas in the combustion chamber, and increase the probability of chemical reactions.Under the assumption that the combustion reaction ends in the spray combustion chamber, the influence of chemical reaction method on the simulation results is significant only when the fuel is excessive.

## Figures and Tables

**Figure 1 materials-14-00657-f001:**
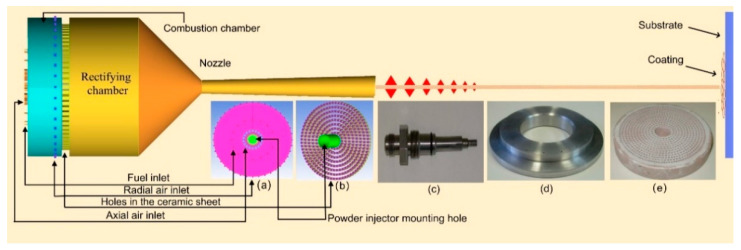
Structural schematic of the activated combustion high-velocity air-fuel (AC-HVAF) spray gun model and the spraying principle: (**a**) left view of the spray gun; (**b**) ceramic sheet and powder injector mounting hole; (**c**) powder injector of actual spray gun; (**d**) propane distributor of actual spray gun; (**e**) porous ceramic sheet of actual spray gun.

**Figure 2 materials-14-00657-f002:**
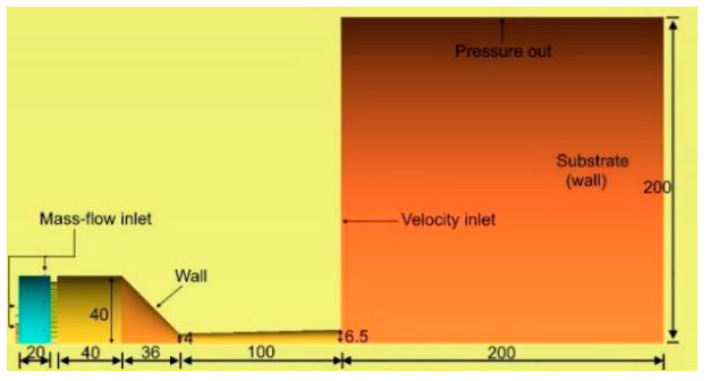
Model size and boundary conditions (unit: mm).

**Figure 3 materials-14-00657-f003:**
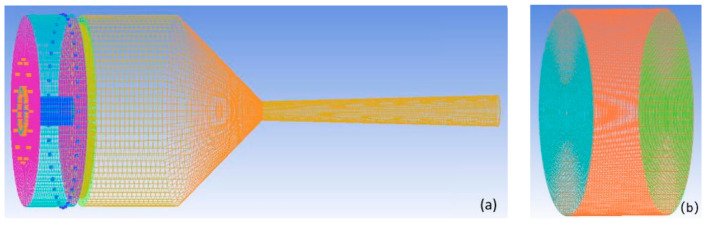
Configuration of the third meshes: (**a**) internal flow field and (**b**) external flow field.

**Figure 4 materials-14-00657-f004:**
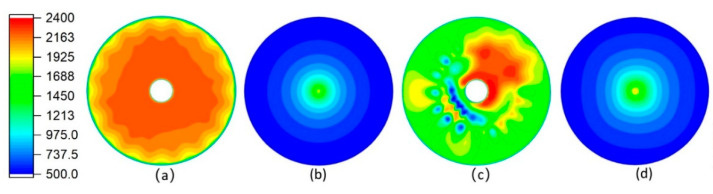
Temperature fields formed using different mesh cells. (**a**,**c**) Temperature field at the middle section of the combustion chamber; (**b**,**d**) temperature field on the surface of the substrate. (**a**,**b**) The fourth meshes are adopted; (**c**,**d**) the first meshes are adopted (the pictures are scaled to different degrees).

**Figure 5 materials-14-00657-f005:**
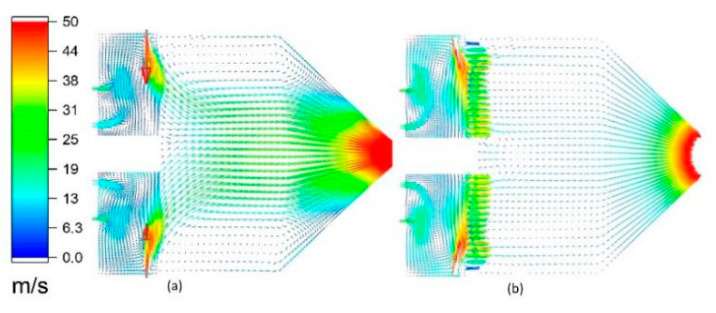
Internal velocity vector diagram of the spray gun: (**a**) second meshes and (**b**) fourth meshes.

**Figure 6 materials-14-00657-f006:**
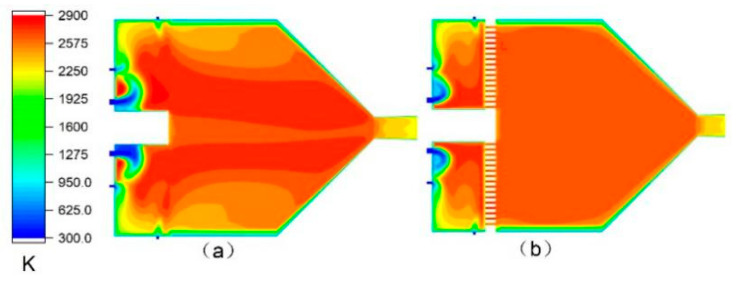
Temperature distribution inside the spray gun: (**a**) second meshes and (**b**) fourth meshes.

**Figure 7 materials-14-00657-f007:**
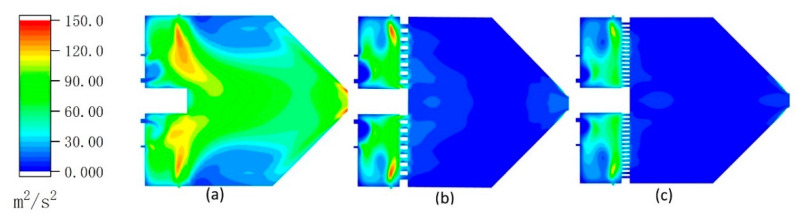
Contours of turbulent kinetic energy inside the spray gun: (**a**) second meshes, (**b**) third meshes, and (**c**) fourth meshes.

**Figure 8 materials-14-00657-f008:**
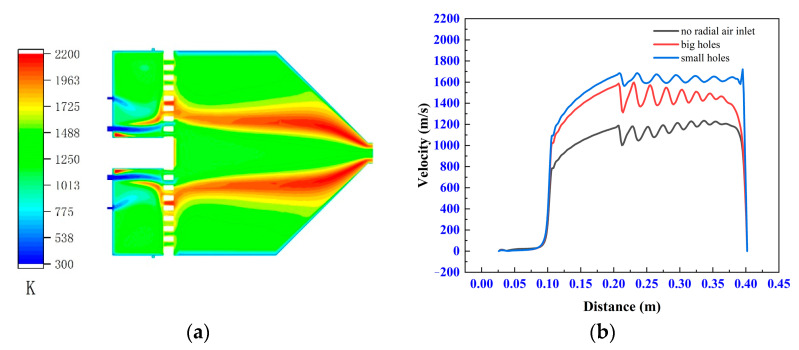
Contours of temperature without radial air inlets (**a**). The velocity at the central axis of the model with no radial inlets, big holes, and small holes in the ceramic sheet (**b**).

**Figure 9 materials-14-00657-f009:**
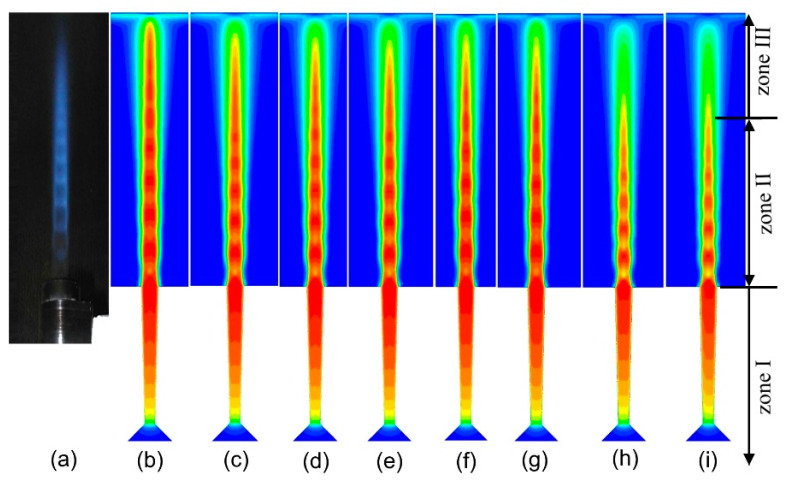
The flame of the actual spray gun and the partial contours of the airflow velocity: (**a**) actual spray gun, (**b**–**i**) experimental scheme corresponding to Table 1.

**Figure 10 materials-14-00657-f010:**
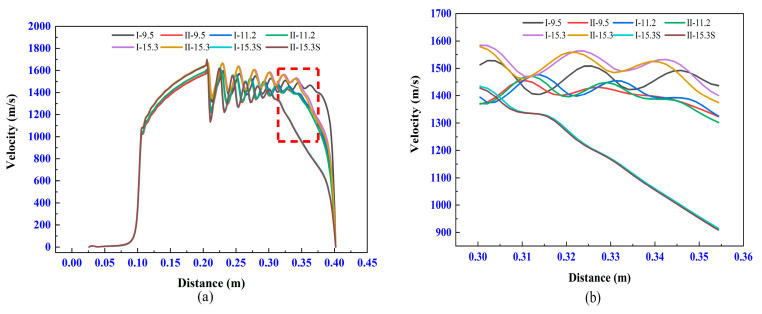
Velocity at the axis of the model (**a**) and a partially enlarged view of the red areas (**b**).

**Figure 11 materials-14-00657-f011:**
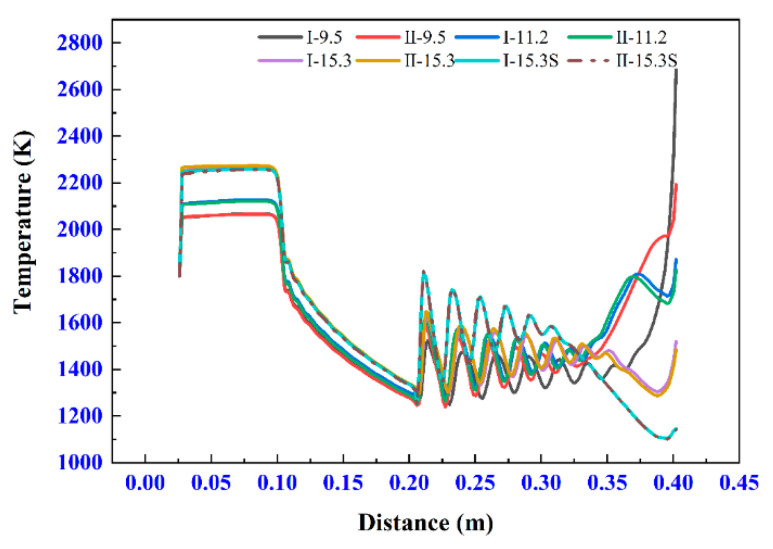
Temperature at the axis of the model.

**Figure 12 materials-14-00657-f012:**
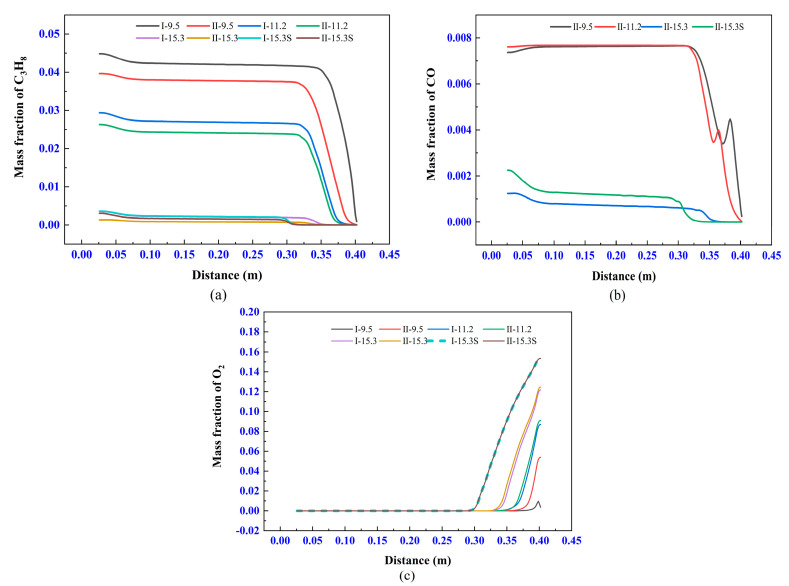
Mass fraction of gas species at the axis of the model: (**a**) C_3_H_8_, (**b**) CO, and (**c**) O_2_.

**Table 1 materials-14-00657-t001:** Four types of the meshes used in the model.

Name of Mesh	With Ceramic Sheet (Yes/No)	Size of Ceramic Holes	Number of Ceramic Holes	Type of Cells	
				Combustion Chamber	Rest of the Model
#1	Yes	2 × 2 mm^2^	452	Tetrahedron	Hexahedron
#2	No	-	-	Hexahedron	Hexahedron
#3	Yes	2 × 2 mm^2^	452	Hexahedron	Hexahedron
#4	Yes	1 × 1 mm^2^	1472	Hexahedron	Hexahedron

**Table 2 materials-14-00657-t002:** Experimental program and parameters.

Experiment Name	Chemical Reaction	Mass Flow Rate at Axial Air Inlets (g/s)	Mass Flow Rate at Radial Air Inlets (g/s)	Mass Flow Rate at Fuel Inlets (g/s)	Air-Fuel Mass Flow Ratio	Total Mass Flow Rate (g/s)
I-9.5	One-step	11.00	24.20	3.70	9.50	38.90
II-9.5	Two-step	11.00	24.20	3.70	9.50	38.90
I-11.2	One-step	8.80	22.60	2.80	11.20	34.20
II-11.2	Two-step	8.80	22.60	2.80	11.20	34.20
I-15.3	One-step	11.00	24.20	2.30	15.30	37.50
II-15.3	Two-step	11.00	24.20	2.30	15.30	37.50
I-15.3-S	One-step	8.80	19.36	1.84	15.30	30.00
II-15.3-S	Two-step	8.00	20.00	1.83	15.30	29.83

**Table 3 materials-14-00657-t003:** Maximum velocity corresponding to different experimental schemes.

**Experiment**	I-9.5	II-9.5	I-11.2	II-11.2	I-15.3	II-15.3	I-15.3S	II-15.3S
**Maximum Velocity (m/s)**	1596	1596	1603	1608	1668	1671	1699	1699

## Data Availability

Data is contained within the article.

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
