# Peer review of "Numerical Analysis of the Activated Combustion High-Velocity Air-Fuel Spraying Process: A Three-Dimensional Simulation with Improved Gas Mixing and Combustion Mode"

_materials, 2021, doi:10.3390/ma14030657_

Round 1

Reviewer 1 Report

Given below are my comments to improvise the quality of the manuscript.

  1. Line 140: Please explain how you chose the range of the velocity.
  2. Line 142: Please mention the said custom function.
  3. Line 143: Gridding or Meshing? I suggest using the word ‘mesh’ instead of ‘grid’.
  4. Line 156: How was it verified that there is almost no difference with the simulation results based on the round holes.
  5. I cannot find the validation of the presented simulation. Please add a suitable validation and further include the accuracy involved in your simulated results.
  6. Line 372: Fuel is excessive to what extent? Kindly quantify (in %).
  7. Please check the English language of your paper using professional language editing software. 

Reviewer 2 Report

In this manuscript, a numerical analysis of the activated combustion high-velocity 2 air-fuel spraying process was performed.

The manuscript is well organized and illustrates the scientific content of the work. This paper is publishable after some minor revisions.
Kindly insert references for all mathematical formulas.

Kindly explain a comparison between the standard k–ε and realizable k–ε turbulence models (advantages and disadvantages) and the accuracy of the standard and realizable k–ε turbulence models should be evaluated and discussed. (references: DOI: 10.1007/s10652-018-9637-1 and so on).

Kindly insert references for “line 209: one-step method and two-step method.”

Kindly insert numerical values for false-diffusion errors (L240: ….false-diffusion errors….,; L243 … supersonic flow field…;  L246:…. simulation accuracy.).

L258: … the reaction probability… Insert mathematical formula and give numerical values for it with the corresponding explanations in the manuscript.

It is known that “Energy in flowing fluid includes the internal energy, potential energy, kinetic energy and the product of its pressure and volume [2]

Zemansky, M.W. Heat and Thermodynamics; McGraw-Hill: New York, NY, USA, 1968.

Kindly insert all mathematical formulas for these energies and if possible, explain deeply in all steps from your numerical analysis.

L267: insert the mathematical formula for the turbulent kinetic energy and explain the corresponding parameters and so on.

Line 369, 371: specify numerical values for: ….. a very large temperature gradient…..,  …. the fuel is excessive….

Minor typos and grammar mistakes. English should be checked once again.

This paper presents an interesting approach and deserved to be published after the mentioned revisions.

Reviewer 3 Report

In general, the paper is well written. The authors present the numerical CFD results using the commercial CFD package Fluent. Results are presented in a concise manner as well as the discussion. 

Some minors comments are listed below:

  1. Please add the definition of Gb by the formula. 
  2. What is the author's opinion if the Gb is not included in the turbulence model?
  3. The authors must briefly explain why a realizable k-epsilon is used as a turbulence model? What is the advantage compare with other models, e.g., k-omega, SST, etc. Or maybe the authors can cite the previous paper regarding the turbulence models in such a system.

Round 2

Reviewer 1 Report

Thank you for taking into account my comments and improvising upon the quality of the manuscript.